# Risk of Secondary Cancer after Adjuvant Tamoxifen Treatment for Ductal Carcinoma In Situ: A Nationwide Cohort Study in South Korea

**DOI:** 10.3390/diagnostics13040792

**Published:** 2023-02-20

**Authors:** Dooreh Kim, Jooyoung Oh, Jeong-Ho Seok, Hye Sun Lee, Soyoung Jeon, Chang Ik Yoon

**Affiliations:** 1Division of Breast Surgery, Department of Surgery, Seoul St. Mary’s Hospital, College of Medicine, The Catholic University of Korea, Seoul 06591, Republic of Korea; 2Department of Psychiatry, Gangnam Severance Hospital, Yonsei University College of Medicine, Seoul 03722, Republic of Korea; 3Biostatistics Collaboration Unit, Yonsei University College of Medicine, Seoul 03722, Republic of Korea

**Keywords:** secondary cancer, tamoxifen, endocrine therapy, ductal carcinoma in situ, DCIS

## Abstract

Endocrine therapy is the mainstay treatment for hormone receptor-positive ductal carcinoma in situ. The aim of this study was to examine the long-term secondary malignancy risk of tamoxifen therapy. The data of patients diagnosed with breast cancer between January 2007 and December 2015 were retrieved from the database of the Health Insurance Review and Assessment Service of South Korea. The International Classification of Diseases, 10th revision, was used to track all-site cancers. Age at the time of surgery, chronic disease status, and type of surgery were considered covariates in the propensity score matching analysis. The median follow-up duration was 89 months. Forty-one patients in the tamoxifen group and nine in the control group developed endometrial cancer. The Cox regression hazard ratio model showed that tamoxifen therapy was the only significant predictor of the development of endometrial cancer (hazard ratio, 2.791; 95% confidence interval, 1.355–5.747; *p* = 0.0054). No other type of cancer was associated with long-term tamoxifen use. In consonance with the established knowledge, the real-world data in this study demonstrated that tamoxifen therapy is related to an increased incidence of endometrial cancer.

## 1. Introduction

Endocrine therapy has substantially reduced the rates of recurrence and death in patients with hormone receptor-positive breast cancer [1]. Tamoxifen, a selective estrogen receptor modulator, is the most widely prescribed breast cancer hormonal therapy agent used for preventive and curative purposes [2]. Tamoxifen has both estrogen agonist and antagonist properties and affects multiple aspects of clinical conditions [3,4]. However, some adverse events of adjuvant tamoxifen treatment can lead to the development of endometrial cancer and pulmonary thromboembolism, which can be fatal [5,6].

Previous studies on the incidence of secondary cancers were primarily focused on the protective effects of tamoxifen in cases of contralateral breast cancer and its adverse outcomes in the treatment of endometrial cancer [7]. The treatment duration and cumulative dose of tamoxifen are positively correlated with the incidence of endometrial cancer [8]; thus, regular surveillance using transvaginal sonography is strongly recommended [9]. However, whether tamoxifen affects the development of other cancers remains unclear. 

Cancer treatment seldom leads to unanticipated harmful consequences of secondary malignancy. Anthracycline is one of the most used chemotherapy agents in breast cancer; it is well known for its relationship with secondary leukemia in patients with breast cancer [10,11,12]. Radiation therapy (RT) is also associated with the development of sarcoma [13]. Radiation-associated sarcoma occurs in the RT field after a long asymptomatic period, and as the RT procedure becomes more precise and articulated, the incidence is expected to decrease [14]. The current study focused on the secondary adverse events of tamoxifen in terms of malignancy while controlling other previously mentioned treatment modalities, which can impact the incidence of certain secondary cancers.

The aim of this study was to examine the risk of all-site secondary cancers, except breast cancer, following adjuvant tamoxifen treatment for ductal carcinoma in situ (DCIS), using real-world, long-term, follow-up data from a national cohort.

## 2. Materials and Methods

The data were extracted from the database of the Health Insurance Review and Assessment Service (HIRA) of South Korea. The HIRA is a national institution responsible for evaluating the credibility of medical expenses in South Korea. During the 15 years from January 2007 to December 2021, the mandatory collection of all patient data on Korean National Health Insurance (NHI) claims was conducted nationwide. The following information on each patient was obtained from the NHI database: demographic information, including personal history; date of disease registration; diagnosis codes; information on procedures; and prescriptions. The International Classification of Diseases, 10th revision (ICD-10) [15], is used to report the diagnoses in the NHI database. The NHI audits the ICD-10 codes and records of prescriptions and procedures regularly to prevent unnecessary medical expenses. The data used for this study were anonymized according to the privacy guidelines of the Health Insurance Portability and Accountability Act in Korea. The study protocol was approved by the Institutional Review Board (IRB) of Seoul St. Mary’s Hospital (local IRB number: KC22ZISI0340). The need for informed consent was waived owing to the retrospective cohort design of the study.

The data of all women aged ≥ 20 years in the NHI database were screened, and those newly diagnosed with DCIS (ICD-10 code, D05) between 2009 and 2015 were selected. To avoid the potential confounding effects of other treatment modalities, patient enrolment was limited to those with in situ disease who presumably had not undergone chemotherapy or targeted therapy. The two-year washout period from 2007 to 2008 excluded patients who had visited a physician for the management of any other type of cancer (ICD-10 code; any C code) prior to the diagnosis of DCIS (D05). Patients are counted at every visit to the hospital; thus, patients who underwent primary curative surgery one year before or after the diagnosis were included to refine and filter duplicates using claims procedure codes. The ICD-10 codes for surgery were as follows: N7133, wide excision; N7134, wide excision of the axillary breast; N7136, wide excision with axillary surgery; N7137, wide excision without axillary surgery; N7138, total mastectomy with axillary surgery; and N7139, total mastectomy without axillary surgery. The day of enrolment for all subjects was the day of DCIS surgery. The follow-up period was calculated from the date of enrolment until the last censored date. The patients were monitored for the occurrence of secondary malignancies until 2021. Patients without any malignancy were censored on 31 December 2021.

The primary outcome was the development of any secondary malignancy, except breast cancer, after adjuvant tamoxifen therapy. Secondary malignancy diagnosis was defined as a cancer diagnosis with the claims ICD diagnostic code C00-97, except for contralateral breast cancer and recurrent/metastatic breast cancer (C50). Patients with testis and prostate cancers (male genital cancer) were also excluded because only women were enrolled in this study. The date of the first claim with the related ICD-10 cancer code was defined as the date of cancer diagnosis. Malignancies with low incidence rates were grouped as “other” (Appendix A). Confounding variables, including chronic comorbidities such as hypertension (I10), diabetes mellitus (E10, E11, E12, E13, and E14), chronic obstructive pulmonary disease (COPD; J44), chronic kidney disease (CKD; N18), liver cirrhosis (K74 and K703), and heart failure (I50), were considered in the process of data collation.

The Student’s t-test was used to compare the continuous variables between the two groups, and the chi-square test or Fisher’s exact test was used to compare the categorical variables. Propensity score matching was performed to minimize selection bias, using the following variables: age at diagnosis, chronic disorders, and type of breast surgery. Propensity score matching was performed using an algorithm in the SAS software (version 9.4, SAS Institute, Cary, NC, USA). The cumulative incidence of each malignancy was estimated using Kaplan–Meier curves and compared using the log-rank test. Cox regression analysis was performed to evaluate the effect of tamoxifen on the development of each secondary malignancy. We utilized the backward likelihood method (entry effects: *p* = 0.05; removal effects: *p* = 0.10). Statistical significance was set at a two-sided *p*-value < 0.05. Statistical analyses were performed using the SAS software. Forest plots were created using Microsoft Excel.

## 3. Results

### 3.1. Patient Cohort

A total of 43,434 patients were diagnosed with DCIS between 2009 and 2015. Of these, 12,032 patients who underwent curative surgery for DCIS within a year after diagnosis were screened for inclusion into this study (Figure 1). Patients with ICD codes for coexisting invasive breast cancer (C05) were excluded, leaving 5021 patients remaining. To exclude patients who had received systemic treatment such as chemotherapy, only patients with DCIS were included into this study. The patients were divided into two groups based on the presence or absence of prescription claims for tamoxifen in their records. A total of 3202 patients who were prescribed tamoxifen within a year after surgery were classified into a tamoxifen group, whereas 1819 patients who did not receive adjuvant endocrine therapy constituted the control group.

### 3.2. Demographic Characteristics and Incidence of Second Primary Cancer

The follow-up period ranged from 23 to 138 months, and the median follow-up period was 89 months. Before matching, the tamoxifen and control groups showed significant differences in the type of breast surgery and in underlying diseases, such as diabetes mellitus and hypercholesterolemia (Table 1). However, the two groups were matched for age, chronic disease status, and type of surgery in a 1:1 ratio. After matching, the demographic characteristics were well balanced between the two groups. The mean age of the patients was 49.7 years. Approximately, 30% of the patients had hypercholesterolemia, whereas 21% had hypertension.

Before matching, 41 of 3205 (1.28%) patients in the tamoxifen group and 9 of 1819 (0.49%) patients in the control group developed endometrial cancer (Table 1). The incidence of endometrial cancer was significantly higher in the group that received adjuvant tamoxifen therapy than in the group that did not (*p* = 0.007). This result remained statistically significant after matching (1.28% in the tamoxifen group and 0.50% in the control group; *p* = 0.013; Table 1). The cumulative incidence of the development of endometrial cancer in the tamoxifen group differed significantly from the incidence of the development of any other type of cancer (Figure 2; log-rank test, *p* = 0.004 and 0.007, before and after matching, respectively). 

The rates of the development of other types of cancer in the two groups were similar. The incidence of all-site cancer was 5.88% (107/1819) and 5.47% (175/3205) in the control and tamoxifen groups, respectively (*p* = 0.537). Thyroid cancer occurred most frequently (88 patients), followed by endometrial cancer (50 patients), ovarian cancer (43 patients), and lung cancer (34 patients) (Table 1).

### 3.3. Risk Factors of Second Primary Cancer

The effect of adjuvant tamoxifen therapy on the incidence of endometrial cancer was analyzed using Cox regression hazard ratio models (Table 2). Univariate analysis of the unmatched cohorts revealed that tamoxifen therapy was the only significant predictor of the development of endometrial cancer (hazard ratio [HR], 2.792; 95% confidence interval [CI], 1.356–5.749; *p* = 0.005; Table 2). Various chronic diseases, such as hypertension, diabetes mellitus, hypercholesterolemia, COPD, CKD, liver failure, and heart failure, were included in the analysis. The results showed that the diseases were not significantly associated with endometrial cancer. The results of the multivariable analysis were consistent with those of the univariate analysis. Analysis of the matched cohorts also showed a strong association between tamoxifen treatment and endometrial cancer in the univariate (HR, 2.782; 95% CI, 1.286–6.021; *p* = 0.009; Figure 3) and multivariate models (HR, 2.672; 95% CI, 1.244–5.739; *p* = 0.012; Appendix A).

The incidence of all-site cancer was associated with older age and hypertension. Before matching, the multivariable model showed that older age and COPD were significant predictive factors for the incidence of all-site cancer (HR, 1.014; 95% CI, 1.001–1.027; *p* = 0.046). In the univariate analysis performed using matched cohorts, only age was an independent predictor of the incidence of all-site cancer (HR, 1.014; 95% CI, 1.001–1.027; *p* = 0.036). The analysis was performed for each type of cancer, and the results showed that tamoxifen was not associated with the risk of developing any other type of cancer (Appendix A). In the multivariate analysis performed using matched cohorts, older age was associated with the risk of a few other types of cancer, including gastric, colorectal, bladder, and lung cancers. 

## 4. Discussion

In this study, analysis of the long-term follow-up data of patients with DCIS retrieved from a national cohort revealed that endometrial cancer is the sole malignancy associated with tamoxifen use. The matched analysis showed that tamoxifen increased the rate of the development of endometrial cancer more than twofold (HR, 2.797; 95% CI, 1.309–5.979; *p* = 0.008). This finding is similar to those of several previous studies [8,16,17,18]. However, the results of the present study showed that there were no statistically significant differences in the incidence and risks of the other secondary malignancies associated with tamoxifen therapy. The Korean national cohort of patients with DCIS demonstrated the long-term safety of adjuvant tamoxifen use in terms of the risk of neoplasms, except endometrial cancer.

Since tamoxifen therapy became the standard of care treatment for hormone receptor-positive breast cancer and hormone receptor-positive DCIS, several studies have been conducted to investigate the neoplastic adverse events associated with tamoxifen use. It is generally recognized that tamoxifen is not associated with the development of secondary cancers other than endometrial cancer. However, the methodologies of the previous studies on this topic may have been limited by confounding variables, such as the inclusion of patients who had received treatment modalities, including chemotherapy, other than tamoxifen therapy. For example, many patients receive anthracycline-based chemotherapy, which puts them at the risk of developing acute myeloid leukemia and myelodysplastic syndrome [19]. The present study was designed to overcome this limitation and re-evaluate the risk of secondary malignancy by using longitudinal data and excluding patients who had received systemic therapy such as chemotherapy and targeted therapy. In addition, the type of surgery was selected as a propensity score matching variable in the present study and was indirectly applied regardless of whether radiotherapy was performed or not.

Rosell et al. recently reported the long-term effects of two and five years of adjuvant tamoxifen therapy on the incidence of second primary cancers in postmenopausal women with early breast cancer [20]. An increased incidence of endometrial cancer was observed in the five-year group, but no additional secondary malignancies were observed. Matsuyama et al. studied the incidence and risks of the development of secondary cancers after tamoxifen treatment in Japanese patients [21]. In their study, the duration of tamoxifen therapy was generally two years or less, and the outcomes were not different between the tamoxifen and non-tamoxifen groups. However, the risk of endometrial cancer owing to tamoxifen use cannot be neglected, especially in young women with a high risk of recurrence who are candidates for the extended use of tamoxifen. Furthermore, the risk of endometrial cancer in the current study is similar to that in previous population-based studies [22,23,24,25].

The national cohort evaluated in the present study comprised relatively young Asian patients who had received no systemic treatment other than tamoxifen. In Asia, breast cancer develops in a bimodal pattern, and the first peak occurs in patients in their 40s and early 50s [26]. The median age of the patients in this study was 49 years, and they had few comorbidities. The present study was designed to assess the independent effect of tamoxifen. In addition, this study had a unique population that was suitable for determining the association between tamoxifen therapy and secondary cancer, after adjusting for age and comorbidities since cancer is a geriatric disease.

The national cohort data on healthcare insurance claims were validated for research purposes using the National Cancer Registry in Korea [27]. Epidemiological studies, especially studies on the incidence of cancer, show highly concordant rates. In addition, the database allows for the collation of a large set of data within a continuum framework, and the operational definition strictly selects the study population.

Our study has several limitations. First, data on the histological type or stage of the secondary cancers were not available; thus, extensive exploration of the study topic was hindered. Second, the insurance claims data lack information regarding lifestyle factors such as diet and smoking history, which can play essential roles as environmental factors. Moreover, the association between the treatment duration and the cumulative dose of tamoxifen and the incidence of cancer was not studied due to practical issues. Finally, this study was conducted using large, longitudinal data on the Korean population. The recent study from South Korea demonstrated the incidence of endometrial disease in breast cancer patients using tamoxifen, and it had a larger sample size because invasive cancer was not excluded [22]. Despite the smaller sample size, our study investigated the isolated effect of tamoxifen in patients with only DCIS who had not received chemotherapy, which is a confounding variable that can affect secondary malignancy. Nonetheless, the findings cannot be generalized to patients from other countries or races.

## 5. Conclusions

In conclusion, this nationwide cohort study of patients with DCIS demonstrated that tamoxifen is not associated with the risk of secondary malignancies, except endometrial cancer, before and after adjusting for age, type of surgery, and comorbidities.

## Figures and Tables

**Figure 1 diagnostics-13-00792-f001:**
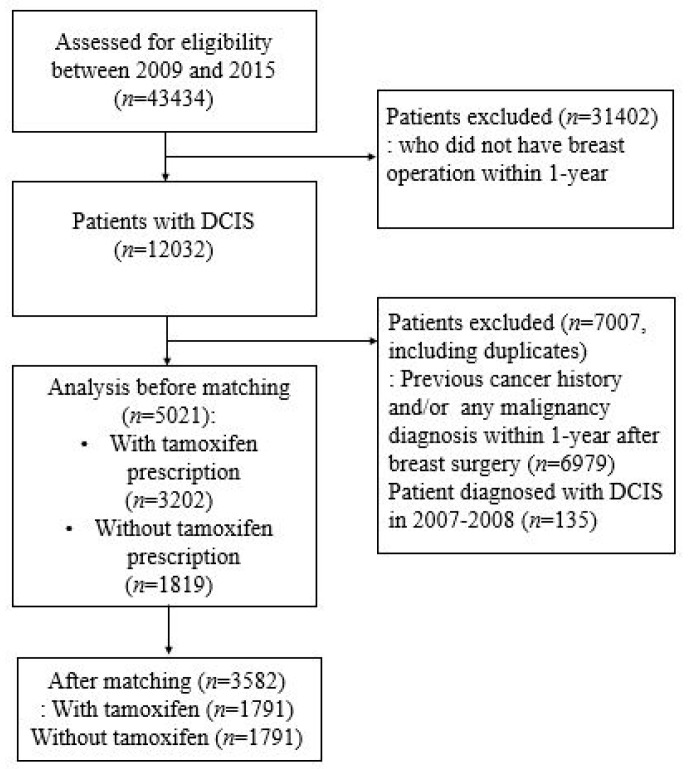
Consort diagram of study design.

**Figure 2 diagnostics-13-00792-f002:**
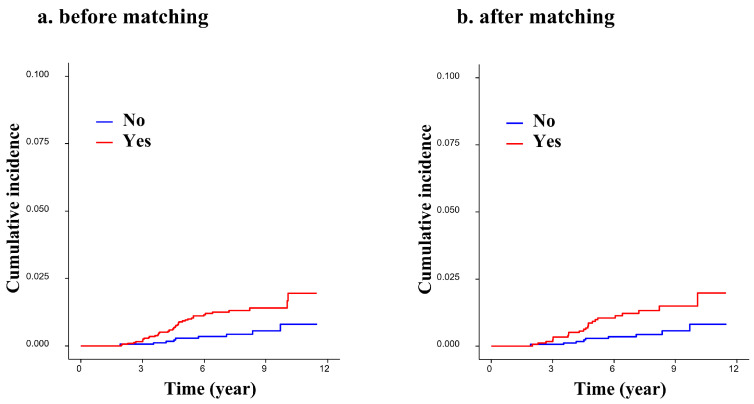
Cumulative incidence of endometrial cancer according to long-term use of tamoxifen: (**a**) before matching, log-rank test, *p*-value = 0.0036; (**b**) after matching, log-rank test, *p*-value = 0.004.

**Figure 3 diagnostics-13-00792-f003:**
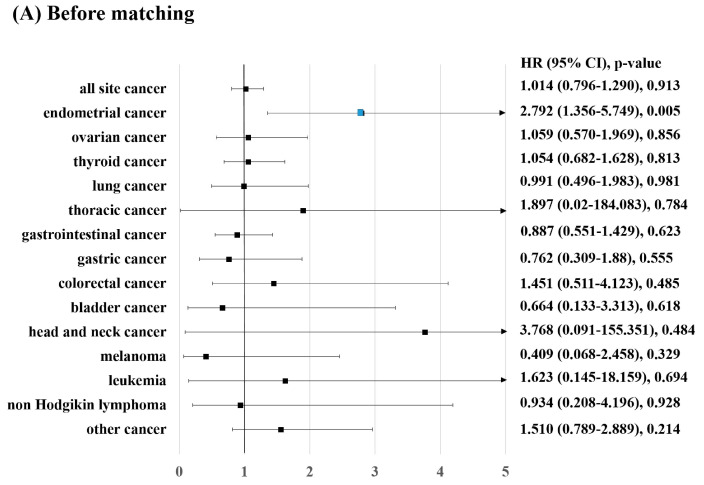
Risk of developing secondary cancer due to long-term tamoxifen use in patients with breast cancer: (**A**) before matching; (**B**) after matching. HR, hazard ratio; CI, confidence interval.

**Table 1 diagnostics-13-00792-t001:** Demographics and incidence of secondary cancer according to long-term tamoxifen use.

	Before Matching				After Matching			
	Total	Tamoxifen		*p*-Value	Total	Tamoxifen		*p*-Value
	*n*(%)	No(*n* = 1819)	Yes(*n* = 3205)		*n*(%)	No(*n* = 1802)	Yes(*n* = 1802)	
All-site cancer								
No	4739(94.38)	1712(94.12)	3027(94.53)	0.537	3385(94.50)	1686(94.14)	1699(94.86)	0.34
Yes	282(5.62)	107(5.88)	175(5.47)		197(5.50)	105(5.86)	92(5.14)	
Endometrial cancer								
No	4971(99.00)	1810(99.51)	3161(98.72)	0.007	3550(99.11)	1782(99.50)	1768(98.72)	0.013
Yes	50(1.00)	9(0.49)	41(1.28)		32(0.89)	9(0.50)	23(1.28)	
Ovarian cancer								
No	4978(99.14)	1803(99.12)	3175(99.16)	0.893	3553(99.19)	1776(99.16)	1777(99.22)	0.852
Yes	43(0.86)	16(0.88)	27(0.84)		29(0.81)	15(0.84)	14(0.78)	
Thyroid cancer								
No	4933(98.25)	1787(98.24)	3146(98.25)	0.978	3528(98.49)	1759(98.21)	1769(98.77)	0.17
Yes	88(1.75)	32(1.76)	56(1.75)		54(1.51)	32(1.79)	22(1.23)	
Lung cancer								
No	4987(99.32)	1806(99.29)	3181(99.34)	0.807	3556(99.27)	1778(99.27)	1778(99.27)	>0.999
Yes	34(0.68)	13(0.71)	21(0.66)		26(0.73)	13(0.73)	13(0.73)	
Thoracic cancer								
No	5020(99.98)	1819(100.00)	3201(99.97)	>0.999	3582(100.00)	1791(100.00)	1791(100.00)	>0.999
Yes	1(0.02)	0(0.00)	1(0.03)		0(0.00)	0(0.00)	0(0.00)	
Gastrointestinal cancer								
No	4951(98.61)	1790(98.41)	3161(98.72)	0.362	3530(98.55)	1763(98.44)	1767(98.66)	0.576
Yes	70(1.39)	29(1.59)	41(1.28)		52(1.45)	28(1.56)	24(1.34)	
Gastric cancer								
No	5002(99.62)	1810(99.51)	3192(99.69)	0.311	3568(99.61)	1782(99.50)	1786(99.72)	0.284
Yes	19(0.38)	9(0.49)	10(0.31)		14(0.39)	9(0.50)	5(0.28)	
Colorectal cancer								
No	5004(99.66)	1814(99.73)	3190(99.63)	0.558	3570(99.66)	1786(99.72)	1784(99.61)	0.563
Yes	17(0.34)	5(0.27)	12(0.37)		12(0.34)	5(0.28)	7(0.39)	
Bladder cancer								
No	5015(99.88)	1816(99.84)	3199(99.91)	0.674	3577(99.86)	1788(99.83)	1789(99.89)	>0.999
Yes	6(0.12)	3(0.16)	3(0.09)		5(0.14)	3(0.17)	2(0.11)	
Head and neck cancer								
No	5019(99.96)	1819(100.00)	3200(99.94)	0.537	3581(99.97)	1791(100.00)	1790(99.94)	>0.999
Yes	2(0.04)	0(0.00)	2(0.06)		1(0.03)	0(0.00)	1(0.06)	
Melanoma								
No	5016(99.90)	1816(99.84)	3200(99.94)	0.359	3577(99.86)	1788(99.83)	1789(99.89)	>0.999
Yes	5(0.10)	3(0.16)	2(0.06)		5(0.14)	3(0.17)	2(0.11)	
Leukemia								
No	5018(99.94)	1818(99.95)	3200(99.94)	>0.999	3580(99.94)	1790(99.94)	1790(99.94)	>0.999
Yes	3(0.06)	1(0.05)	2(0.06)		2(0.06)	1(0.06)	1(0.06)	
Non-Hodgkin lymphoma								
No	5014(99.86)	1816(99.84)	3198(99.88)	0.709	3578(99.89)	1788(99.83)	1790(99.94)	0.625
Yes	7(0.14)	3(0.16)	4(0.12)		4(0.11)	3(0.17)	1(0.06)	
Other cancer								
No	4977(99.12)	1806(99.29)	3171(99.03)	0.354	3553(99.19)	1778(99.27)	1775(99.11)	0.576
Yes	44(0.88)	13(0.71)	31(0.97)		29(0.81)	13(0.73)	16(0.89)	
Type of surgery								
BCS	4482(89.27)	1584(87.08)	2898(90.51)	<0.001	3132(87.44)	1564(87.33)	1568(87.55)	0.84
mastectomy	539(10.73)	235(12.92)	304(9.49)		450(12.56)	227(12.67)	223(12.45)	
DM								
No	4415(87.93)	1630(89.61)	2785(86.98)	0.006	3222(89.95)	1608(89.78)	1614(90.12)	0.739
Yes	606(12.07)	189(10.39)	417(13.02)		360(10.05)	183(10.22)	177(9.88)	
HTN								
No	3929(78.25)	1437(79.00)	2492(77.83)	0.333	2834(79.12)	1421(79.34)	1413(78.89)	0.742
Yes	1092(21.75)	382(21.00)	710(22.17)		748(20.88)	370(20.66)	378(21.11)	
Hyperlipidemia								
No	3564(70.98)	1335(73.39)	2229(69.61)	0.005	2626(73.31)	1312(73.26)	1314(73.37)	0.94
Yes	1457(29.02)	484(26.61)	973(30.39)		956(26.69)	479(26.74)	477(26.63)	
COPD								
No	4894(97.47)	1771(97.36)	3123(97.53)	0.709	3486(97.32)	1746(97.49)	1740(97.15)	0.535
Yes	127(2.53)	48(2.64)	79(2.47)		96(2.68)	45(2.51)	51(2.85)	
CKD								
No	4974(99.06)	1798(98.85)	3176(99.19)	0.226	3551(99.13)	1776(99.16)	1775(99.11)	0.857
Yes	47(0.94)	21(1.15)	26(0.81)		31(0.87)	15(0.84)	16(0.89)	
Liver cirrhosis								
No	5005(99.68)	1810(99.51)	3195(99.78)	0.095	3572(99.72)	1785(99.66)	1787(99.78)	0.526
Yes	16(0.32)	9(0.49)	7(0.22)		10(0.28)	6(0.34)	4(0.22)	
Heart failure								
No	4975(99.08)	1798(98.85)	3177(99.22)	0.182	3550(99.11)	1775(99.11)	1775(99.11)	>0.999
Yes	46(0.92)	21(1.15)	25(0.78)		32(0.89)	16(0.89)	16(0.89)	
Age (mean ± SD)	49.746 ± 10.224	49.396 ± 10.787	49.944 ± 9.886	0.075	49.385 ± 10.501	49.413 ± 10.681	49.357 ± 10.321	0.8736

BCS, breast-conserving surgery; DM, diabetes; HTN, hypertension; COPD, chronic obstructive pulmonary disease; CKD, chronic kidney disease; SD, standard deviation.

**Table 2 diagnostics-13-00792-t002:** Univariable and multivariable analysis of endometrial cancer risk factors.

	Before Matching				After Matching			
	Univariable Model		Multivariable Model		Univariable Model		Multivariable Model	
	HR(95% CI)	*p*-Value	HR(95% CI)	*p*-Value	HR(95% CI)	*p*-Value	HR(95% CI)	*p*-Value
Tamoxifen								
No	ref		Ref		ref		ref	
Yes	2.792(1.356–5.749)	0.005	2.664(1.309–5.421)	0.007	2.782(1.286–6.021)	0.0094	2.672(1.244–5.739)	0.0118
Type of surgery								
BCS	ref		Ref		ref		ref	
mastectomy	1.165(0.496–2.734)	0.725	1.281(0.558–2.940)	0.559	1.340(0.516–3.480)	0.547	1.446(0.573–3.649)	0.434
DM								
No	ref		Ref		ref		ref	
Yes	1.454(0.682–3.098)	0.332	1.094(0.490–2.443)	0.827	1.342(0.470–3.826)	0.583	1.414(0.481–4.152)	0.529
HTN								
No	ref		Ref		ref		ref	
Yes	1.548(0.846–2.835)	0.157	1.213(0.594–2.476)	0.596	1.273(0.572–2.833)	0.555	1.318(0.529–3.284)	0.554
Hyperlipidemia								
No	ref		Ref		ref		ref	
Yes	1.629(0.919–2.888)	0.095	1.387(0.727–2.644)	0.321	0.997(0.447–2.222)	0.994	0.925(0.392–2.180)	0.858
COPD								
No	ref		Ref		ref		ref	
Yes	0.837(0.116–6.065)	0.86	0.994(0.193–5.105)	0.994	0.592(0.035–10.079)	0.717	0.515(0.035–7.482)	0.627
CKD								
No	ref		Ref		ref		ref	
Yes	2.323(0.321–16.827)	0.404	2.947(0.576–15.085)	0.195	1.882(0.110–32.149)	0.662	1.475(0.082–26.403)	0.792
Liver cirrhosis								
No	ref		Ref		ref		ref	
Yes	3.211(0.193–53.523)	0.416	2.244(0.145–34.822)	0.563	5.868(0.344–100.088)	0.222	6.691(0.440–101.865)	0.171
Heart failure								
No	ref		Ref		ref		ref	
Yes	2.258(0.312–16.354)	0.42	3.005(0.608–14.865)	0.177	1.726(0.101–29.487)	0.706	1.547(0.112–21.295)	0.744
Age	1.017(0.991–1.044)	0.209	1.007(0.976–1.039)	0.666	1.008(0.975–1.041)	0.644	1.006(0.968–1.045)	0.768

HR, hazard ratio; CI, confidence interval; BCS, breast-conserving surgery; DM, diabetes; HTN, hypertension; COPD, chronic obstructive pulmonary disease.

## Data Availability

The data presented in this study are available on request from the corresponding author.

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
