# Peer review of "Risk of Secondary Cancer after Adjuvant Tamoxifen Treatment for Ductal Carcinoma In Situ: A Nationwide Cohort Study in South Korea"

_diagnostics, 2023, doi:10.3390/diagnostics13040792_

Round 1
Reviewer 1 Report
This article evaluated the risk of secondary cancer after adjuvant tamoxifen treatment for ductal carcinoma in situ, this study was conducted based on a nationwide cohort in South Korea. By reviewing 43,434 DCIS patients underwent curative surgery without chemotherapy. They matched 1891 patients in tamoxifen group and without tamoxifen group, respectively. The cox regression hazard ratio model showed that tamoxifen therapy was the only predictor for the development of endometrial cancer. The statistical method was solid, the expression of the results was clear. However, this area was already well study and well learned, even in South Korea, several studies had already reported that premenopausal Korean women with breast cancer who received tamoxifen as adjuvant hormone therapy had a significantly increased risk of endometrial hyperplasia, polyps, carcinoma, and other uterine cancers compared with those who were not treated with adjuvant hormone therapy. Their sample size was even larger, please see the reference. Therefore, although this study was well conducted, the novelty was still not enough to be published in Diagnostic.
Ki-Jin Ryu , Min Sun Kim, Ji Yoon Lee, et al. Risk of Endometrial Polyps, Hyperplasia, Carcinoma, and Uterine Cancer After Tamoxifen Treatment in Premenopausal Women With Breast Cancer. JAMA Netw Open . 2022 Nov 1;5(11):e2243951.
Author Response
Thank you for the thorough review. We are aware of the recent publication about the association between tamoxifen and endometrial disease. Their sample size was much larger because they included all patients with invasive breast cancer. In our study, we investigated the isolated effect of tamoxifen in patients with DCIS who did not receive chemotherapy, which is a confounding variable that can affect secondary malignancy. Despite the small sample size, we believe this article conveys the association between tamoxifen and other types of secondary cancer, which are not explored frequently elsewhere.

Reviewer 2 Report
In the submitted manuscript Kim et al. showed that endometrial cancer was the only type of cancer associated with long-term tamoxifen use in South Korean hormone receptor (HR)-positive ductal carcinoma in situ (DCIS) breast cancer patients, and that tamoxifen therapy was the only significant predictor of the development of that type of cancer.
There are several both minor and major drawbacks which must be corrected or improved before this manuscript is suitable for publishing:
1) Lines 37-38: Authors wrote “The treatment duration and cumulative dose of tamoxifen are positively correlated with the incidence of endometrial cancer” so it is unclear why they didn’t also include those variables in their study.
2) Table 1 needs improvements:
- Precisely state what those numbers present. I suppose ‘number of cases’ (n) and ‘percentage’ (%), while for age ‘mean ± standard deviation’.
- It is unclear for which statistical tests are those p-values. I suppose chi-square or exact test, and t-test for age. If yes, that must be mentioned in ‘Methods’.
- There is no need to present mean age and its SD with such many decimals. In addition, age is usually presented with median and range (or IQR). Furthermore, if parametric t-test was used, where is the proof that age follows normal distribution?
- “HR, hazard ratio; CI, confidence interval” was mentioned in footnotes while HR and CI were not presented in that table!
3) On Figure 2 provide number of samples (n) for each group and p-value for log-rank test.
4) It is unclear which software was used for creating Figure 3, and all abbreviations presented on it must be explained in figure legend.
5) In multivariable analysis are usually included only variables that are statistically significant in univariable analysis. Otherwise, variables that become statistically significant only in multivariable analysis should be commended, like for several cases in your supplementary tables!
6) Supplementary Figure 1 was not provided for reviewing!
7) Supplementary tables need improvements:
- Name of table must come before table.
- All abbreviations mentioned in table must be explained in footnotes.
- It must be explained what those yellow cells present.
- Supplementary Table 5 is missing lots of data, like information for “Heart failure” for univariable model before matching, while results for “After matching” analyses are completely missing.
8) Minor improvements:
Line 31: “agonistic” instead “agonist”
Line 65: It is unclear what “target-agent therapy” means; I suppose “targeted therapy”.
Lines 74-75: Precisely state what was considered as an endpoint of follow up time.
Line 166: “Supplementary Tables 1-14” instead “Supplementary Table”
Author Response
In the submitted manuscript Kim et al. showed that endometrial cancer was the only type of cancer associated with long-term tamoxifen use in South Korean hormone receptor (HR)-positive ductal carcinoma in situ (DCIS) breast cancer patients, and that tamoxifen therapy was the only significant predictor of the development of that type of cancer.
There are several both minor and major drawbacks which must be corrected or improved before this manuscript is suitable for publishing:
1) Lines 37-38: Authors wrote “The treatment duration and cumulative dose of tamoxifen are positively correlated with the incidence of endometrial cancer” so it is unclear why they didn’t also include those variables in their study.
Response: Thank you for your valuable comment. We agree that the study of treatment duration and cumulative dose of tamoxifen would have made the research much more comprehensive. However, further analysis is currently not possible without new data extraction because extracted data from the HIRA database is permanently deleted 1 year after the study ends. A new analysis requires a wait of 12–18 months due to latency. The same dataset is not available to be opened; therefore, we could not perform any further analysis.
2) Table 1 needs improvements:
- Precisely state what those numbers present. I suppose ‘number of cases’ (n) and ‘percentage’ (%), while for age ‘mean ± standard deviation’.
- It is unclear for which statistical tests are those p-values. I suppose chi-square or exact test, and t-test for age. If yes, that must be mentioned in ‘Methods’.
- There is no need to present mean age and its SD with such many decimals. In addition, age is usually presented with median and range (or IQR). Furthermore, if parametric t-test was used, where is the proof that age follows normal distribution?
- “HR, hazard ratio; CI, confidence interval” was mentioned in footnotes while HR and CI were not presented in that table!
Response: Thank you for the detailed review. We revised Table 1 per your suggestions. We have also added relevant information in the Methods. Currently, we do not have data on the median value and range of age, and the database cannot be re-opened. Therefore, we assumed that the age followed a normal distribution based on the “law of large numbers” and central limit theorem.
Methods:
“Student’s t-test was used to compare continuous variables between the two groups, and the chi-square test or Fisher’s exact test was used to compare categorical variables.”
3) On Figure 2 provide number of samples (n) for each group and p-value for log-rank test.
Response: Thank you for the valuable comment. We have added the P values for log-rank test in the figure legends. The number of samples can be found in Table 2.
Figure 2 Legend
“Figure 2. Cumulative incidence of endometrial cancer according to long-term use of tamoxifen. (A) before matching, log-rank test, p-value=0.0036, (B) after matching, log-rank test, p-value=0.004.”
4) It is unclear which software was used for creating Figure 3, and all abbreviations presented on it must be explained in figure legend.
Response: Thank you for the comment. Figure 3 was created using Microsoft Excel. We have stated this in the Methods. All abbreviations in the figure have been defined in the figure legend.
Methods:
“Figure 3 was created using Microsoft Excel.”
Figure 3 Legend
“Figure 3. Risk of developing secondary cancer due to long-term tamoxifen use in breast cancer patients. (A) before matching, (B) after matching; HR, hazard ratio; CI, confidence interval”
5) In multivariable analysis are usually included only variables that are statistically significant in univariable analysis. Otherwise, variables that become statistically significant only in multivariable analysis should be commended, like for several cases in your supplementary tables!
Response: Thank you for your valuable comment. We applied the backward likelihood method (significance level for entering effect = 0.05 and removing effects = 0.05). Because the backward likelihood method was used, all variables were included and analyzed in this study.
6) Supplementary Figure 1 was not provided for reviewing!
Response: Thank you for your valuable comment. We have added supplementary figure 1.
7) Supplementary tables need improvements:
- Name of table must come before table.
- All abbreviations mentioned in table must be explained in footnotes.
- It must be explained what those yellow cells present.
- Supplementary Table 5 is missing lots of data, like information for “Heart failure” for univariable model before matching, while results for “After matching” analyses are completely missing.
Response: Thank you for your valuable comments. Supplementary tables are all updated following comments.
8) Minor improvements:
Line 31: “agonistic” instead “agonist”
Line 65: It is unclear what “target-agent therapy” means; I suppose “targeted therapy”.
Lines 74-75: Precisely state what was considered as an endpoint of follow up time.
Line 166: “Supplementary Tables 1-14” instead “Supplementary Table”
Response: Thank you for your valuable comments. There were typos as you pointed out. Per your suggestion, we have corrected them.

Reviewer 3 Report
Thank you for the opportunity to review the manuscript “Risk of secondary cancer after adjuvant tamoxifen treatment for ductal carcinoma in situ: a nationwide cohort study in South 3 Korea”.
The effect of tamoxifen to increase the risk of ovarian cancer is well known.
The article respond correctly to the problem state in the title.
It will increase the quality of the paper if the authors report the duration of tamoxifen treatment and cumulative dose, the proportion of patients receiving radiotherapy (which may have an important effect of the incidence of secondary cancers) and number of patients with the conservative surgery.
The article is well written and needs small English adjustments.
Mean must be followed by SD.
Introduction is quite small and may be improved.
In the discussions the authors must give more detail why the incidence of endometrial is much higher than the one reported in the literature.
Another problem is the related by the statistical analysis (HR = 1 but the statistical significance is reached?)
" COPD 160 were significant predictive factors for the incidence of all-site cancer (HR, 1.014; 95% CI, 161 1.001-1.027; p=0.046). In the univariate analysis performed using matched cohorts, only 162 age was an independent predictor of the incidence of all-site cancer (HR, 1.014; 95% CI, 163 p=0.036)".
The references are a little updated and in small number.
Author Response
Thank you for the opportunity to review the manuscript “Risk of secondary cancer after adjuvant tamoxifen treatment for ductal carcinoma in situ: a nationwide cohort study in South 3 Korea”.
The effect of tamoxifen to increase the risk of ovarian cancer is well known.
The article respond correctly to the problem state in the title.
It will increase the quality of the paper if the authors report the duration of tamoxifen treatment and cumulative dose, the proportion of patients receiving radiotherapy (which may have an important effect of the incidence of secondary cancers) and number of patients with the conservative surgery.
Response: Thank you for reviewing our manuscript. We agree that the study of treatment duration and cumulative dose of tamoxifen would have made the research much more comprehensive. However, further analysis is currently not possible without new data extraction because extracted data from the HIRA database is permanently deleted 1 year after the study ends. A new analysis requires a wait of 12–18 months due to latency. The same dataset is not available to be opened; therefore, we could not perform any further analysis.
In South Korea, radiation therapy following breast-conserving surgery is the standard treatment for breast cancer. We assumed that all patients with BCS had received RT. This explains why we used the type of surgery as a variable when performing the propensity score matching. A recent study suggests that RT does increase the risk of sarcoma, but at a very low risk (1). The purpose of this study was to identify the risk of secondary malignancy according to endocrine treatment (tamoxifen); radiation has already been reported in other studies. In addition, radiation can act as a confounding variable; therefore, we adjusted the surgical method (total mastectomy vs. breast-conserving surgery).
- Snow A, Ring A, Struycken L, Mack W, Koç M, Lang JE. Incidence of radiation induced sarcoma attributable to radiotherapy in adults: A retrospective cohort study in the SEER cancer registries across 17 primary tumor sites. Cancer Epidemiol. 2021 Feb;70:101857. doi: 10.1016/j.canep.2020.101857. Epub 2020 Nov 26.
The article is well written and needs small English adjustments.
Response: Thank you for this remark. We have re-edited this revised article and and attached the proofreading certificate with this re-submission.
Mean must be followed by SD.
Response: Thank you for your comments. We have added the relevant values (mean±SD) in Table 1
Introduction is quite small and may be improved.
Response: Thank you for your comment. We have added information to the Introduction as follows:
Introduction
“Cancer treatment seldom leads to unanticipated harmful consequences of secondary malignancy. Anthracycline is one of the most used chemotherapy agents in breast cancer; it is well-known for its relationship with secondary leukemia in patients with breast cancer. Radiation therapy (RT) is also associated with the development of sarcoma. Radiation-associated sarcoma occurs in the RT field after a long asymptomatic period, and as the RT procedure becomes more precise and articulated, the incidence is expected to decrease. The current study focused on the secondary adverse events of tamoxifen in terms of malignancy while controlling other aforementioned treatment modalities, which can impact the incidence of certain secondary cancers.”
In the discussions the authors must give more detail why the incidence of endometrial is much higher than the one reported in the literature.
Response: Thank you for your comment. The incidence of endometrial cancer is within the range of known risks in previously published articles.
Discussion:
“This finding is similar to those of several previous studies [8, 11-13].”
- Neven P, Vernaeve H. Guidelines for monitoring patients taking tamoxifen treatment. Drug Saf. 2000;22(1):1-11.
- Braithwaite RS, Chlebowski RT, Lau J, George S, Hess R, Col NF. Meta-analysis of vascular and neoplastic events associated with tamoxifen. J Gen Intern Med. 2003;18(11):937-47.
- Davies C, Pan H, Godwin J, Gray R, Arriagada R, Raina V, et al. Long-term effects of continuing adjuvant tamoxifen to 10 years versus stopping at 5 years after diagnosis of oestrogen receptor-positive breast cancer: ATLAS, a randomised trial. Lancet. 2013;381(9869):805-16.
- Cuzick J, Sestak I, Cawthorn S, Hamed H, Holli K, Howell A, et al. Tamoxifen for prevention of breast cancer: extended long-term follow-up of the IBIS-I breast cancer prevention trial. Lancet Oncol. 2015;16(1):67-75.
Another problem is the related by the statistical analysis (HR = 1 but the statistical significance is reached?)
Response: We have read through the manuscript and could not locate where we stated a HR of 1. We would appreciate it if the reviewer could specifically point out where the problem is.
" COPD 160 were significant predictive factors for the incidence of all-site cancer (HR, 1.014; 95% CI, 161 1.001-1.027; p=0.046). In the univariate analysis performed using matched cohorts, only 162 age was an independent predictor of the incidence of all-site cancer (HR, 1.014; 95% CI, 163 p=0.036)".
Response: Thank you for your comment. We apologize for the non-clarity of the text related to COPD; we have made relevant revisions.
Results:
“After adjusting for other variables, the multivariable model showed that older age and COPD were significant predictive factors for the incidence of all-site cancer (HR, 1.014; 95% CI, 1.001-1.027; p=0.046).”
> “Before matching, adjusting for other variables, the multivariable model showed that older age and COPD were significant predictive factors for the incidence of all-site cancer (HR, 1.014; 95% CI, 1.001-1.027; p=0.046).”
The references are a little updated and in small number.
Response: Thank you for your comments. We have added and updated the reference according to your comment.

Round 2
Reviewer 2 Report
Authors have satisfactorily answered to all my questions and adequately improved quality of this manuscript.
Author Response
Thank you very much for the review
Reviewer 3 Report
All the sugestions have been amended.
Author Response
Thank you very much for the thorough review.